# Predictors of death among severe COVID-19 patients admitted in Hawassa City, Sidama, Southern Ethiopia: Unmatched case-control study

**Samuel Misganaw**[1]*, **Betelhem Eshetu**[1], **Adugnaw Adane**[2], **Tarekegn Solomon**[1]

1 Department of Epidemiology, School of Public Health, College of Medicine and Health Sciences, Hawassa University, Hawassa, Ethiopia, 2 Department of Human Physiology, Faculty of Medical Sciences, College of Medicine and Health Sciences, Hawassa University, Hawassa, Ethiopia

* Samimisganew@gmail.com

## Abstract

### Introduction

Since COVID-19 was announced as a worldwide pandemic, the world has been struggling with this disease. In Ethiopia, there is some information on the epidemiological characteristics of the disease and treatment outcomes of COVID-19 patients. But, there is limited evidence related to predictors of death in COVID-19 patients.

### Objective

To assess the predictor of death among severely ill COVID-19 patients admitted in Hawassa city COVID-19 treatment centers.

### Methods

An institution-based unmatched case-control study was conducted at Hawassa city COVID-19 treatment centers from May 2021 to June 2021. All severe COVID-19-related deaths from May 2020 to May 2021 were included in the case group whereas randomly selected discharged severe COVID-19 patients were included in the control group. Extracted information was entered into Epi-data 4.6 and exported to SPSS 25 for analysis. Multivariable binary logistic regression was run to assess predictors. The result was presented as an adjusted odds ratio with a 95% confidence interval. Variables with a 95% confidence interval which not included one were considered statistically significant.

### Result

A total of 372 (124 cases and 248 controls) patients were included in the study. Multivariable analysis revealed age $\geq$ 65 years (AOR = 2.62, 95% CI = 1.33–5.14), having shortness of breath (AOR = 1.87, 95% CI = 1.02–3.44), fatigue (AOR 1.78, 95% CI = 1.09–2.90), altered consciousness (AOR 3.02, 95% CI = 1.40, 6.49), diabetic Mellitus (AOR = 2.79, 95% CI = 1.16–6.73), chronic cerebrovascular disease (AOR = 2.1, 95% CI = 1.23, 3.88) were found to be predictors of death.

**Data Availability Statement:** The data used for this study cannot be shared publicly due to the inclusion of sensitive participant information. Data is available upon request from Fanuel Belayneh

(Postgraduate Program Coordinator, School of Public Health, Hawassa University) via email (fanuelbelayneh@gmail.com) for researchers who meet the criteria for access to confidential data.

**Funding:** The author(s) received no specific funding for this study.

**Competing interests:** All Authors declared that no conflict of interest exist.

**Abbreviations:** AOR, Adjusted Odds Ratio; BSC, Bachelor of Science; CI, Confidence Interval; COPD, Chronic Obstructed Pulmonary Disease; COVID-19, Corona Virus Disease 2019; EPHI, Ethiopian Public Health Institute; GCS, Glasgow-Coma Scale; HIV/AIDS, Human Immunodeficiency Virus/ Acquired Immunodeficiency Syndrome; IQR, Interquartile Range; PT-PCR, Reverse Transcription Polymerase Chain Reaction; RNA, Ribonucleic Acid; SARS-COV-2, Severe Acute Respiratory Syndrome Corona Virus 2; SOB, Shortness of Breath; SPSS, Statistical Package for Social Sciences; $SPO_2$, Peripheral Oxygen Saturation; WHO, World Health Organization.

## Conclusion

Older age, shortness of breath, fatigue, altered consciousness, and comorbidity were predictors of death in Severe COVID-19 patients.

## Introduction

Coronavirus disease (COVID-19) is an infectious disease caused by a single-stranded RNA virus called severe acute respiratory syndrome coronavirus 2 (SARS-CoV-2) [1]. In December 2019, a cluster of patients with pneumonia of unknown cause emerged in Wuhan, China [2]. On January 2020, severe acute respiratory syndrome coronavirus 2 (SARS-CoV-2) was identified as a causative agent for that observed pneumonia cases [3].

World health organization named the disease caused by SARS-CoV-2 "COVID-19" (coronavirus disease 2019). On 11 March 2020, COVID-19 spreads to 144 countries and more than 118,000 cases and 4,000 deaths were reported; and WHO announced it as a pandemic [4]. In Africa, Egypt reported the first COVID-19 case on 14 February followed by Algeria on 25 February 2020 [5, 6], and Ethiopia reported the first COVID-19 case on 13 March 2020 [7].

Even if precautionary measures were practiced to prevent the rapid spread of the disease, the virus was quickly expanding all over the world and causing millions of death. As of the WHO report on 28 September 2021, there have been over 231 million cases and more than 4.7 million deaths worldwide. The United States of America reported the highest number of cases and deaths followed by India [8]. The case fatality rate of the disease caused by SARS-CoV-2 is 3.26–4.16% in Latin America; 5.8% in the United States [9]. In Africa, a total of 5,998,863 cases and 144,957 deaths were reported [8]. As of the EPHI report on 19 September 2021, there were over 332 thousand confirmed COVID-19 cases and more than 5 thousand deaths in Ethiopia [10].

Since COVID-19 was announced as a worldwide pandemic, the world has been struggling with this disease [4], but the pandemic is still ongoing and the number of confirmed cases and mortality rates are changing every day. Virus mutations and appearing of new variants are also challenging to public health as they are more contagious and cause more severe illnesses [11, 12]. Similar to other RNA viruses, SARS-CoV-2 also continually mutates and new variants are appearing [11]. A variant of interest is defined as an isolate of SARS-CoV-2 that has genotypic and/or phenotypic changes compared to the reference genome. A variant of concern is defined as a variant of interest that has evidence of one or more increases in transmissibility or detrimental change in COVID-19 epidemiology, increase in virulence or change in clinical disease presentation; decrease in the effectiveness of available diagnostics, vaccines, therapeutics or public health and social measures [12, 13].

Globally, cases of the Alpha variant have been reported in 193 countries, while 142 countries have reported cases of the Beta variant; 96 countries have reported cases of the Gamma variant and the Delta variant has been reported in 187 countries [8]. In Ethiopia Alpha, Beta, and Delta SARS-COV-2 variants were detected [10].

Based on the WHO report on 28 September 2021, over 3.3 million new cases and over 55,000 new deaths were reported globally during the week of 20–26 September 2021. The African region reported over 87,000 new cases and over 2,500 new deaths in the reported week. The number of deaths in Africa shows a 5% increase as compared to the previous week [8]. In Ethiopia, currently, people neglect the benefit of social distancing, hand washing, staying at home, using a mask, and suffering from the critical phase of COVID-19 [14]. As of the EPHI report on 19 September 2021, Ethiopia was ranked in the fourth position by the number of

confirmed cases and in the sixth position by the number of deaths due to COVID-19 in Africa. A total of 9,857 new cases and 201 new deaths were reported during the week of 13–19 September 2021. The weekly number of cases and death increased by 21% compared with the past week. The rate of positivity was higher in Sidama region compared with other regions which was 36% [10].

The severity of the disease and treatment outcome of the patients varies from person to person. It may range from asymptomatic infection to severe disease with complications and lastly even death [15, 16]. It displayed a wide spectrum of clinical signs and symptoms, which included: fever, cough, sore throat, nasal congestion, sputum, headache, diarrhea, fatigue, dyspnea, chest tightness, myalgia, nausea, rhinorrhea, dizziness or confusion, hemoptysis, anorexia, vomiting, chest and abdominal pain [17].

After the first case of COVID-19 disease were reported in China the virus spread around the world and cause millions of deaths. Following that, international studies identified potential predictors of mortality among COVID-19 patients [18–20].

Regarding disease severity and predictor of mortality, studies identified different potential risk factors for disease severity and mortality among COVID-19 patients; including older age, male sex, pre-existing comorbidities, patient's vital sign, and clinical symptoms during admission, radiographic and laboratory findings [19, 21–25].

Advanced age, male sex, cardiovascular comorbidities, acute cardiac or kidney injury, and lymphocytopenia, hypertension, diabetes, COPD, and history of CVD), acute organ injury, conferred an increased risk of death [26–28]. Age ⏥65 years was identified as a strong predictor of death for COVID-19 patients [20, 21, 29, 30]. Males were more likely to develop severe disease with complications and had much higher mortality than females [19, 22, 31–33]. Related to comorbidity status, hypertension, diabetes [20], chronic kidney disease, stroke [34], cardiovascular or cerebrovascular disease [21], and liver disease [18] were identified as predictors of death among severe COVID-19 patients.

Patient's vital signs and presenting symptoms on admission such as SpO2 less than 90%, RR greater than 20 breath /minute, heart rate greater than 100peats/ minute systolic blood pressure less than 90 mmHg, dyspnea, cough, breathing difficulty, vomiting and consciousness disorder were identified as a predictor of death [19, 24, 35, 36].

Identification of predictors of death and understanding the characteristic of severe COVID-19 patients is very important to provide efficient, equitable, and appropriate management for COVID-19 patients. In Africa, there are limited data regarding regional predictors of mortality among COVID -19 patients [33].

In Ethiopia, there is some information on epidemiological characteristics of the disease, Knowledge, attitudes, and practices of the population related to the COVID-19 pandemic, and treatment outcome of COVID-19 patients. But there is limited evidence related to predictors of death in COVID-19 in Ethiopia including Hawassa.

To fill the gap it is important to conduct more research on the area of identifying risk factors for disease severity and predictor of death among COVID-19 patients in our local setting. Therefore, the purpose of this study was to assess the predictors of death in severe COVID-19 patients in Hawassa, southern Ethiopia in 2021.

## Method and materials

### Study setting and period

This study was conducted at Hawassa city COVID-19 treatment centers. Severe COVID-19 patients admitted in the centers from May 2020 to May 2021 were included in the study and data was extracted from 30 May 2021 to 30 June 2021.

Hawassa city is located 273 km south of Addis Ababa, Ethiopia. During the pandemic, there was one quarantine center (at Hawassa University), two isolation centers (at pyramid hotel and Dukale wakayo hotel), and two treatment centers (Mehal sub-city and Hawassa University Comprehensive Specialized Hospital).

The COVID-19 treatment center located in Mehal sub-city has 100 beds including 8 critical beds and two mechanical ventilators. Hawassa University Compressive Specialized Hospital COVID-19 treatment center has 100 beds including six intensive care unit beds of which four of them are equipped with mechanical ventilation.

## Study design

An institution-based unmatched case-control study.

## Population

**Source population.** All patients with a confirmed diagnosis of severe COVID-19 using PT-PCR and admitted to COVID-19 treatment centers in Hawassa city.

**Study population.** All patients admitted to Hawassa city COVID-19 treatment centers with a confirmed diagnosis of severe COVID-19 from May 2020 to May 2021.

*Case.* Cases were patients admitted to Hawassa city COVID-19 treatment centers from May 2020 to May 2021 with a confirmed diagnosis of severe COVID-19 and whose treatment outcome was death.

*Control.* Controls were patients admitted to Hawassa city COVID-19 treatment centers from May 2020 to May 2021 with a confirmed diagnosis of severe COVID-19 and who recover from the disease and were discharged.

## Inclusion and exclusion criteria

**Inclusion criteria.** All severe COVID-19 patients who were on treatment and follow-up at the Centers during the study period and whose treatment outcome status is known as dead or discharged were included in the study.

**Exclusion criteria.** Severe COVID-19 patients whose chart is incomplete for age, sex, comorbidity, vital sign, and clinical symptom were excluded.

Severe COVID-19 patients whose outcome status is unknown due to transfer to other hospitals or any other reason that resulted in the discharge of the patient before the observation of the outcome were excluded from the study.

## Sample size determination

The sample size was calculated via online Open Epi-info from the findings of previous similar studies. The calculation was done for four potential determinants (age ≥65, cough, male sex, and oxygen saturation (spo2) ≤89%) which were consistently significant in many studies. To determine the appropriate sample size; two-sided confidence level, the power of the study, the ratio of controls to cases, the proportion of controls with exposure, relatively least extreme odds ratio to be detected, and lastly, 10% for the incomplete chart was added. Based on this, the maximum calculated sample size was 366 (122 cases and 244 controls) and it was considered the minimum sample size of the study [36] (Table 1).

*Sample of cases.* All confirmed severe COVID-19-related death in the study setting from May 2020 to May 2021 and whose chart is completed related to age, sex, comorbidity, and presenting signs and symptoms.

**Table 1. Sample size calculation for assessment of predictor of death among severely ill COVID-19 patients admitted to Hawassa city COVID-19 treatment centers, Hawassa, Sidama, Ethiopia, 2021.**

| Variables | CI | Power | AOR | The ratio of control to case | The proportion of control exposed | case | control | 10% | Total | Ref. |
|---|---|---|---|---|---|---|---|---|---|---|
| Age ≥65 | 95% | 80% | 3.765 | 2 | 30.4 | 33 | 66 | 10 | 109 | [21] |
| Cough | 95% | 80% | 2.06 | 2 | 25.7 | 111 | 222 | 33 | 366 | [36] |
| SPO2 ≤89% | 95% | 80% | 2.959 | 2 | 25.7 | 49 | 98 | 15 | 162 | [19] |
| Male sex | 95% | 80% | 2.876 | 2 | 51.4 | 54 | 108 | 16 | 178 | |

*Sample of controls.* From all severe COVID-19 patients who were discharged with recovery from the COVID-19 treatment centers from May 2020 to May 2021, the sample of the control group was selected using a 1:2 case-to-control ratio.

## Sampling technique and procedure

From May 2020 to May 2021, a total of 1,032 confirmed COVID-19 patients were admitted to COVID-19 treatment centers in the study area. Of the total admitted patients, 673 patients were severe cases. These severe COVID-19 patients are divided into the case (dead) and control (discharged) groups based on their outcome status. Of the total 673 severe COVID-19 patients, 128 were dead and 545 were discharged. Among the dead, 4 incomplete charts were excluded and 124 patients were included in the study. From the discharge, the sample of control was selected using a 1:2 case-to-control ratio. Finally, 372 (124 cases and 248 controls) were included in the study.

To select the sample of control, a sampling frame was developed using discharged patient's medical registration number. Then simple random sampling technique was used to select the required number of controls from each COVID-19 treatment center. The number of controls selected from each setting was proportional to the number of selected cases (Fig 1).

## Variables

**Dependent variables.**  Treatment outcome of COVID-19 patients (Death versus Discharge).

## Independent variables

**Demographic variables**; Age, sex, place of residence

**Signs and symptoms during admission;** Temperature, heart rate, respiratory rate, blood pressure, fever, cough, sore trout, shortness of breath, chest pain, headache, loss of appetite, fatigue, altered consciousness, joint pain, and vomiting.

**Laboratory findings;** White blood cell count, hemoglobin, hematocrit, platelets count, creatinine, urea, sodium, potassium, aspartate transaminase, alkaline phosphatase.

**Comorbidity;** Hypertension, diabetes, cardiovascular disease, cerebrovascular disease, asthma, chronic obstructed pulmonary disease, chronic kidney disease, chronic liver disease, neurological disease, cancer, hematologic disease, malnutrition, HIV/AIDS, and tuberculosis.

## Data collection technique

Data were extracted from the patient's card using a standardized data collection tool, which was a modified version of the WHO/ International Severe Acute Respiratory and Emerging Infection Consortium (ISARIC) case record form for severe acute respiratory infection clinical characterization [37]. The extracted information about the patients were including demographic data, pre-existing comorbidities, vital signs, clinical symptoms, signs during admission, and the patient's outcome status.

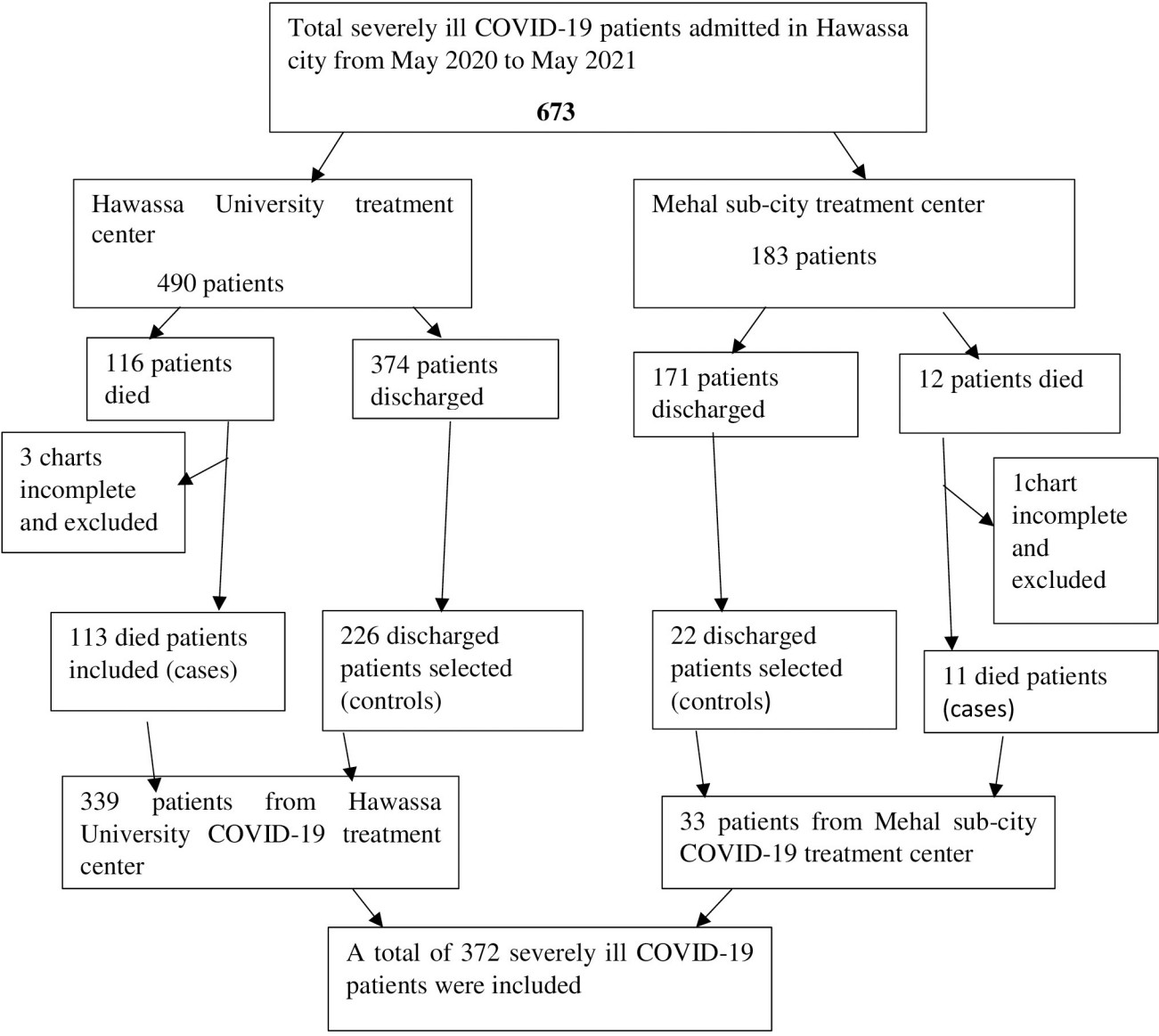

**Fig 1. Sampling technique for assessment of predictor of death among severely ill COVID-19 patients admitted to Hawassa city COVID-19 treatment centers, Hawassa, Sidama, Ethiopia, 2021.**

### Data quality and management

Amendment to the tool was made after a pre-test on the data extraction tool conducted on 19 randomly selected charts at shashamene treatment center. Training on the data collection tool was given to four BSC nurse data collectors and one supervisor. Data consistency and completeness were checked before data entry and analysis.

### Data management and statistical analysis

The data were coded, entered, cleaned, and stored in epi-data version 4.6 and exported to SPSS version 25 software for analysis. Descriptive statistics were presented by frequency and percentage for categorical variables and median and interquartile range for continuous

variables. The statistically significant difference between died and discharged groups in terms of independent variables was assessed by Pearson's chi-square test for categorical variables and Continuous variables were assessed by the Mann-Whitney U test due to their non-normally distribution. Those variables with p-value < 0.05 was defined as having statistical significance difference in terms of COVID-19 outcome. Multivariable binary logistic regression was run to assess the association between the dependent variable and independent variables. Initially, bivariate analysis was conducted and variables having p-value < 0.25 were further included in multivariate analysis to determine independent risk factors associated with COVID-19 mortality. For the final model, model assumption such as multicollinearity and Hosmer and Lemeshow goodness of fit test was checked and the result of the final model was presented as an adjusted odds ratio with a 95% confidence interval. Variables having a 95% confidence interval that does not include one in the final model were considered independent predictors of COVID-19 death.

## Operational definitions

**Sever COVID-19 disease. Adolescent or adult** with clinical signs of pneumonia (fever, cough, dyspnea, fast breathing) **plus one of the following:** Respiratory rate $\geq$ 30 breaths/min;

Peripheral oxygen saturation (SpO2) < 90% on room air, arterial partial pressure of oxygen (PaO2)/fraction of inspired oxygen (FiO2) $\leq$ 300mmHg (1mmHg = 0.133 kPa).

**A child** with clinical signs of pneumonia (cough or difficulty in breathing) **plus at least one** of the following: Central cyanosis or SpO2 < 90%, Severe respiratory distress (e.g. fast breathing, grunting, very severe chest indrawing), general danger sign (inability to breastfeed or drink, lethargy or unconsciousness, or convulsions), fast breathing (in breaths/min): < 2 months: $\geq$ 60; 2–11 months: $\geq$ 50; 1–5 years: $\geq$ 40 [38].

**COVID-19 death.** A death due to COVID-19 is defined for surveillance purposes as a death resulting from a clinically compatible illness, in a probable or confirmed COVID-19 case, unless there is a clear alternative cause of death that cannot be related to COVID disease (e.g. trauma). There should be no period of complete recovery from COVID-19 between illness and death [39].

## Ethical considerations

Ethical approval was obtained from the Hawassa University, College of Medicine and Health Sciences Institutional Review Board. Besides, permission letters were given to respective institutions in the study area, (Hawassa University comprehensive specialized Hospital, Mehal sub city COVID-19 treatment center). Moreover, permission was obtained from the responsible person of each study institute after explaining the purpose and significance of the study. Face masks, sanitizer and gloves were provided for data collectors. The patient data was taken anonymously by using the patient's medical registration number and the personal information of the patients stayed confidential.

## Results

### Socio-demographic characteristics and clinical presentation

A total of 372 severely ill COVID-19 patients were included in the study (124 cases and 248 controls). The age of the patients ranged from 1 year to 90 years with a median of 46 (IQR: 25) years. The majority 233 (62.6%) of the patients were male. Compared between the two groups, the patients in the death event group had a higher male proportion (70.2% vs. 58.9%), and older age (median 58 vs. 42) than patients who are discharged. In addition, the case group was

more likely to present with fever (49.6% vs. 41.1%), shortness of breath (80.6% vs. 68.1%), fatigue (51.6% vs. 37%), and altered consciousness (21% vs. 6.5%) compared to control group (Table 2).

**Table 2. Socio-demographic characteristics and presenting symptoms of severe COVID-19 patients in Hawassa city COVID-19 treatment centers, Hawassa, Sidama, Ethiopia, 2021.**

| Variables | Case (n = 124) | Control (n = 248) | p-value |
|---|---|---|---|
| **Median (IQR) age (Years)** | 58(28) | 42(22) | 0.001 |
| **Age groups (Years)** | | | |
| <65 | 91(73.4%) | 225(90.7%) | 0.001 |
| ≥65 | 33(26.6%) | 23(9.3%) | |
| **Sex** | | | |
| Male | 87(70.2%) | 146(58.9%) | 0.034 |
| Female | 37(29.8%) | 102(41.1%) | |
| **Residence** | | | |
| Rural | 35 (28.2%) | 69 (27.8%) | 0.935 |
| Urban | 89 (71.8%) | 179 (72.2%) | |
| **Fever** | | | |
| Yes | 51 (41.1%) | 123 (49.6%) | 0.123 |
| No | 73 (58.9%) | 125 (50.4%) | |
| **Cough** | | | |
| Yes | 111 (89.5%) | 222 (89.5%) | 1.00 |
| No | 13 (10.5%) | 26 (10.5%) | |
| **Sore throat** | | | |
| Yes | 8 (6.5%) | 10 (4.0%) | 0.305 |
| No | 116 (93.5%) | 238 (96.0%) | |
| **Shortness of breath** | | | |
| Yes | 100 (80.6%) | 169 (68.1%) | 0.011 |
| No | 24 (19.4%) | 79 (31.9%) | |
| **Chest Pain** | | | |
| Yes | 36 (29.0%) | 86 (34.7%) | 0.274 |
| No | 88 (71.0%) | 162 (65.3%) | |
| **Headache** | | | |
| Yes | 38 (30.6%) | 96 (38.7%) | 0.127 |
| No | 86 (69.4%) | 152 (61.3%) | |
| **Loss of appetite** | | | |
| Yes | 37 (29.8%) | 68 (27.4%) | 0.625 |
| No | 87 (70.2%) | 180 (72.6%) | |
| **Fatigue** | | | |
| Yes | 64 (51.6%) | 92 (37.1%) | 0.007 |
| No | 60 (48.4%) | 156 (62.9%) | |
| **Altered consciousness** | | | |
| Yes | 26 (21.0%) | 16 (6.5%) | 0.001 |
| No | 98 (79.0%) | 232 (93.5%) | |
| **Joint pain** | | | |
| Yes | 12 (9.7%) | 35 (14.1%) | 0.225 |
| No | 112 (90.3%) | 213 (85.9%) | |
| **Vomiting** | | | |
| Yes | 21(16.9) | 31(12.5) | 0.245 |
| No | 103(83.1) | 217(87.5) | |

## Baseline vital signs and laboratory findings

On admission, the median respiratory rate was 32 breaths per minute (IQR: 8) and the median heart rate was 120 beats per minute (IQR: 24 beats/min). The patients in the case group had higher respiratory rate (median 35.8 vs. 30 beats/min) and heart rate (median 105 vs. 98 beats/min). On the laboratory findings, patients in the case group had a lower value of Hematocrit (median 39.8 vs. 41.15), platelets count (median 234 vs. 225), and higher white blood cell count (median 13 vs. 8.4), and urea (median 46.5 vs. 29.6) than the control group (Table 3).

## Comorbidity status of the patients

Of the total participants, 223 (59.9%) patients presented with one or more comorbidity. The proportion of patients presenting with comorbidity was higher in the case group (79% vs. 50%) than in the controls. Diabetes mellitus (37.1% vs. 17.7%), hypertension (35.5% vs. 20.2%), and cerebrovascular disease (17.7% vs. 4.4%) were noted with a higher proportion in the case group than the control group (Table 4).

## Predictors of death among patients with severe COVID-19

The bivariate logistic regression analysis was initially performed on socio-demographic variables, presenting symptom and comorbidity parameters of the patients. As a result, variables including male sex, age $\geq$ 65 years, shortness of breath, fatigue, altered consciousness, Diabetic Mellitus, hypertension, and cerebrovascular disease showed significant predictors of death events.

To determine independent risk factors associated with COVID-19 mortality, a multivariable analysis was performed. Variables that have a p-value of < 0.25 in bivariate analysis were

**Table 3. Baseline vital signs and laboratory findings of severe COVID-19 patients in Hawassa city COVID-19 treatment centers, Hawassa, Sidama, Ethiopia, 2021.**

| Variables | Case | Control | p-value |
|---|---|---|---|
| **Presenting vital sign** | | | |
| Median (IQR) body temperature in ˚C | 37.2 (1.2) | 37.1 (1.2) | 0.412 |
| Median (IQR) heart rate per minute | 105 (86) | 98 (22) | 0.003 |
| Median (IQR) respiratory rate per minute | 35.81(16) | 30 (6) | 0.001 |
| Median (IQR) systolic blood pressure | 122.5(25) | 120 (22) | 0.407 |
| Median (IQR) diastolic blood pressure | 73.5(18) | 73(14) | 0.947 |
| **Complete cell count** | | | |
| Median (IQR) White blood cell count (x $10^3$/L) | 13 (10) | 8.4 (6.6) | 0.001 |
| Median (IQR) Hemoglobin (g/dL) | 12.65 (4.2) | 13.2(2.9) | 0.052 |
| Median (IQR) Hematocrit (%) | 39.8(12.9) | 41.15(8.8) | 0.179 |
| Median (IQR) platelets count (x $10^3$/L) | 234(147) | 225(133.5) | 0.85 |
| **Renal function test** | | | |
| Median (IQR) creatinine (mg/dl) | 1.08(0.8) | 0.91(0.37) | 0.001 |
| Median (IQR) urea (mg/dl) | 46.5(30.4) | 29.6(20) | 0.001 |
| **Electrolyte** | | | |
| Median (IQR) sodium (mEq/L) | 135 (10) | 134(7.7) | 0.17 |
| Median (IQR) potassium (mEq/L) | 4.3(1.6) | 3.9(0.87) | 0.039 |
| **Liver function test** | | | |
| Median (IQR) Aspartate transaminase (IU/L) | 47.65(34.65) | 36(31) | 0.051 |
| Median (IQR) Alkaline phosphatase (IU/L) | 139 (141) | 106 (103) | 0.154 |

**Table 4. Comorbidity status of severe COVID-19 patients at Hawassa city COVID-19 treatment centers, Hawassa, Sidama, Ethiopia, 2021.**

| Variable | Case (n = 124) # (%) | Control (n = 248) # (%) | p-value |
|---|---|---|---|
| **At least one comorbidity** | | | |
| Yes | 98 (79.0) | 125 (50.4) | 0.001 |
| No | 26 (21.0) | 123 (49.6) | |
| **Hypertension** | | | |
| Yes | 44 (35.5) | 50 (20.2) | 0.001 |
| No | 80 (64.5) | 198 (79.8) | |
| **Cardiovascular disease/non-hypertension/** | | | |
| Yes | 15 (12.1) | 24 (9.7) | 0.473 |
| No | 109 (87.9) | 224 (90.3) | |
| **Diabetes Mellitus** | | | |
| Yes | 46 (37.1) | 44 (17.7) | 0.001 |
| No | 78 (62.9) | 204 (82.3) | |
| **Cerebrovascular disease** | | | |
| Yes | 22 (17.7) | 11(4.4) | 0.001 |
| No | 102 (82.3) | 237 (95.6) | |
| **Asthma** | | | |
| Yes | 7 (5.6) | 15 (6.0) | 0.876 |
| No | 117 (94.4) | 233 (94) | |
| **Obstructive pulmonary disease/non-Asthma/** | | | |
| Yes | 1 (3.2) | 4 (0.4) | 0.026 |
| No | 120 (96.8) | 247 (99.6) | |
| **Kidney disease** | | | |
| Yes | 3 (2.4) | 7 (2.8) | 0.821 |
| No | 121 (97.6) | 241 (97.2) | |
| **Liver disease** | | | |
| Yes | 2 (1.6) | 3 (1.2) | 0.750 |
| No | 122 (98.4) | 245 (98.8) | |
| **Neurological disease** | | | |
| Yes | 3 (2.4) | 1 (0.4) | 0.076 |
| No | 121 (97.6) | 247 (99.6) | |
| **Cancer** | | | |
| Yes | 3 (2.4) | 1 (0.4) | 0.076 |
| No | 121 (97.6) | 247 (99.6) | |
| **Hematologic disease/non-cancer/** | | | |
| Yes | 6 (4.8) | 3 (1.2) | 0.032 |
| No | 118 (95.2) | 245 (98.8) | |
| **Malnutrition** | | | |
| Yes | 3 (2.4) | 6 (2.4) | 1.000 |
| No | 121 (97.6) | 242 (97.6) | |
| **HIV/ADIS** | | | |
| Yes | 7 (5.6) | 5 (2.0) | 0.062 |
| No | 117 (94.4) | 243 (98.0) | |
| **Tuberculosis** | | | |
| Yes | 8 (6.5) | 7 (2.8) | .093 |
| No | 116 (93.5) | 241 (97.2) | |

selected for multivariate analysis. Multicollinearity effect was not observed (variance inflation factor (VIF) value lies between 1 and 2 in each involved variable). The final model was checked for goodness of fit using Hosmer and Lemeshow test and the value was 0.422 which means the actual model and the expected one has no significant difference.

Based on multivariate analysis results age ≥ 65 years, shortness of breath, fatigue, altered consciousness, diabetic Mellitus, and chronic cerebrovascular disease were the independent predictors of COVID-19 death.

The risk of mortality among severe COVID-19 patients with age ≥ 65 years was 2.62 times higher as compared to the patients with age < 65 years (AOR = 2.62, 95%CI = 1.33–5.14).

Patients who presented with shortness of breath were 87% more likely to have died than a patient without shortness of breath (AOR = 1.87, 95% CI = 1.02–3.44). Mortality among COVID-19 patients who presented with a symptom of fatigue was 78% higher compared to patients without fatigue (AOR = 1.78, 95% CI = 1.09–2.90). Altered consciousness also increased the risk of COVID-19 death by 3 times (AOR = 3.02, 95% CI = 1.40–6.49).

The risk of mortality among COVID-19 patients with cerebrovascular disease was 2.79 times higher than patients without cerebrovascular disease (AOR = 2.79, 95%CI = 1.16–6.73). Patients with diabetics were two times as high as the patients without diabetics to die due to COVID-19 disease (AOR = 2.18, 95% CI = 1.23–3.88) (Table 5).

## Discussion

This study assessed the predictors of death in severe COVID-19 patients who were admitted at Hawassa city COVID-19 treatment centers. Age ≥ 65 years, shortness of breath, fatigue, altered consciousness, diabetic Mellitus, and chronic cerebrovascular diseases were the independent predictors of COVID-19 death.

The finding of this study revealed that older age was associated with a higher risk of death among severe COVID-19 patients. Patients with age ≥ 65 years had higher odds of death as compared to patients with age < 65 years. Similar to this finding, studies in the USA reported older age was found as a potential risk factor for death among COVID-19 patients [18, 26, 27]. A cohort study in china also reveals that age ⬚ 65 years was a strong predictor of death for COVID-19 patients [21]. A study conducted in Bangladesh also explained that mortality among hospitalized COVID-19 patients with age> 65 years was 3.59 times more likely higher as compared to the patients with age < 65 years [20]. This is due to there is correlation between age and natural immunity; as natural immunity declines gradually at older ages. As age increases, decreases the production of T cells and immunological responses to pathogens. It leads to less body fitness to fight infection and increased vulnerability and susceptibility to adverse health outcomes or death when exposed to infection [40]. Older people are also vulnerable to adverse drug reactions which may either reduce organ function at an older age or taking multiple drugs due to comorbidities [41, 42].

The result of this study also found that having shortness of breath at admission is a significant factor that predicts a death outcome. Patients who had a history of SOB at presentation had a higher risk of death outcome than those who had no history of SOB. Similarly, studies conducted in Ethiopia [43], Nigeria, and Congo reported shortness of breath as a significant risk factor for death among COVID-19 patients [24, 25, 36]. This is due to that severe pneumonia can cause significant gas exchange disturbances and lead to hypoxemia. Hypoxia reduces the energy production required for cell metabolism and increases the body's anaerobic digestion. Acidosis and oxygen free radicals accumulated in the cell destroy the phospholipid layer of the cell membrane. As hypoxia continues, the intracellular calcium ion concentration increases significantly, leading to a series of cell damage processes [44]. As a result, shortness

**Table 5. Bivariable and multivariable logistic regression result of severe COVID-19 patients, Hawassa, Sidama, Ethiopia, 2021.**

| Variable | Case (n = 124) # (%) | Control (n = 248) # (%) | COR(95%CI) | AOR(95%CI) |
|---|---|---|---|---|
| **Age** | | | | |
| <65 | 91(73.4) | 225(90.7) | 1 | 1 |
| ≥65 | 33(26.6) | 23(9.3) | 3.55 (1.98–6.37) | **2.62 (1.33–5.14)**\* |
| **Sex** | | | | |
| Male | 87(70.2) | 146(58.9) | 1.64 (1.04–2.6) | 1.19 (0.71–1.99) |
| Female | 37(29.8) | 102(41.1) | 1 | 1 |
| **Fever** | | | | |
| Yes | 51 (41.1) | 123 (49.6) | 0.71 (0.46–1.1) | 0.77 (0.45–1.28) |
| No | 73 (58.9) | 125 (50.4) | 1 | 1 |
| **Shortness of breath** | | | | |
| Yes | 100 (80.6) | 169 (68.1) | 1.95 (1.16–3.27) | **1.87 (1.02–3.44)**\* |
| No | 24 (19.4) | 79 (31.9) | 1 | 1 |
| **Headache** | | | | |
| Yes | 38 (30.6) | 96 (38.7) | 0.7 (0.44–1.11) | 0.75 (0.45–1.25) |
| No | 86 (69.4) | 152 (61.3) | 1 | 1 |
| **Fatigue** | | | | |
| Yes | 64 (51.6) | 92 (37.1) | 1.81 (1.17–2.8) | **1.78 (1.09–2.9)**\* |
| No | 60 (48.4) | 156 (62.9) | 1 | 1 |
| **Altered consciousness** | | | | |
| Yes | 26 (21) | 16 (6.5) | 3.85 (1.98–7.49) | **3.02 (1.4–6.49)**\* |
| No | 98 (79.0) | 232 (93.5) | 1 | 1 |
| **Joint pain** | | | | |
| Yes | 12 (9.7) | 35 (14.1) | 0.65 (0.33–1.31) | 0.59 (0.27–1.29) |
| No | 112 (90.3) | 213 (85.9) | 1 | 1 |
| **Vomiting** | | | | |
| Yes | 21(16.9) | 31(12.5) | 1.43 (0.79–2.6) | 1.47 (0.75–2.91) |
| No | 103(83.1) | 217(87.5) | 1 | 1 |
| **Hypertension** | | | | |
| Yes | 44 (35.5) | 50 (20.2) | 2.18(1.35–3.52) | 1.29 (0.71–2.34) |
| No | 80 (64.5) | 198 (79.8) | 1 | 1 |
| **Cerebrovascular disease** | | | | |
| Yes | 22 (17.7) | 11(4.4) | 4.65 (2.17–9.94) | **2.79 (1.16–6.73)**\* |
| No | 102 (82.3) | 237 (95.6) | 1 | 1 |
| **Diabetic Mellitus** | | | | |
| Yes | 46 (37.1) | 44 (17.7) | 2.73 (1.68–4.46) | **2.18 (1.23–3.88)**\* |
| No | 78 (62.9) | 204 (82.3) | 1 | 1 |
| **Tuberculosis** | | | | |
| Yes | 8 (6.5) | 7 (2.8) | 2.37 (0.84–6.71) | 2.9 (0.89–9.38) |
| No | 116 (93.5) | 241 (97.2) | 1 | 1 |

\* = significantly associated, AOR = Adjusted odds ratio, COR = crude odds ratio

of breath is a manifestation of decreased lung function and is considered a sign of a life-threatening condition.

In addition, fatigue was identified as a significant risk factor for death among severe Covid -19 patients in this study. In the same way, a study conducted in china explained patients presented with fatigue were at a 20% higher risk of death than those without fatigue. However, no

significant relationships were found between mortality and fever, cough, diarrhea, headache, abdominal pain, dizziness, nausea, and chest pain in this study as well as in the previous study [23].

This study also identified that altered consciousness was a risk factor for mortality among COVID-19 patients. The patient presented with altered consciousness were three times high risk to die than those without altered consciousness. Similarly, a study conducted in china suggested that there was a direct link between consciousness impairment and death in COVID-19 patients. Patients whose Glasgow coma scale core was less than 9 were at high risk of death than patients with a GCS score >14 [45]. This may be because altered consciousness is a manifestation of organ failure due to continued hypoxemia.

In the present analysis, diabetes was found to be an important predictor of death among severe COVID-19 patients. Patients with diabetics had higher odds of death than patients without diabetics. This result is consistence with other previous studies [20, 22]. A study conducted in Bangladesh explained patients with diabetics were 1.87 times higher than patients without diabetes to die [20]. A study conducted in Addis Ababa, Ethiopia also showed death in severe COVID-19 patients is found to be associated with being diabetic [43]. This could be because diabetes mellitus, especially if poorly controlled, lead to compromised immunity that decreases the body's ability to fight off any infection [46]. They are more susceptible to being infected by viruses, bacteria, and fungi than individuals without diabetes. Also, the chances of having or developing another chronic illness are higher than in non-diabetic individuals [46, 47]. As a result, these patients might be at an increased risk of SARS-COV-2 infection which could result in a worse disease prognosis.

This study identified risk of death among COVID-19 patients with cerebrovascular disease was higher than without the disease. In a previous study, a history of cerebrovascular disease is associated with a 2.78-fold increased risk of mortality compared to patients without underlying cerebrovascular disease [48]. A cohort study in china also consistently explained the risk of death among COVID-19 patient with cerebrovascular disease were 2.4 times higher than patients without the disease [21].

## Conclusion

This study tried to assess the predictors of death among severe COVID-19 patients admitted to Hawassa treatment centers. The study used a relatively larger sample size compared with a study done in Addis Ababa. It also addresses many variables and it is the study in the southern Ethiopian district which may include sociocultural differences from other study areas. Majorities of studies on predictors of death have been conducted in the context of Asian and American countries. In Africa, there are limited data regarding regional predictors of mortality among COVID -19 patients including Ethiopians and this finding may be one input. In conclusion, this study found that age was an important demographic variable that predict death outcomes among severely ill COVID-19 patients. Mortality among severe COVID-19 patients with age ≥65 years was higher as compared to the patients with age < 65 years. Clinical signs and symptoms such as shortness of breath, fatigue, and altered consciousness were the most significant predictor of death in severe COVID-19 patients. Regarding pre-existing comorbidities, having diabetes and cerebrovascular disease at admission were significant predictors to have death outcomes among severely ill COVID-19 patients.

## Recommendations

Policymakers and those responsible to develop COVID-19 triage protocols shall give more focus on those risk groups and should be done with a more sensitive triaging method to pick

them. If severe COVID-19 patients present with older age, difficulty in breathing, fatigue, altered consciousness, and comorbidities like diabetes and cerebrovascular disease, it is necessary to be alert for further deterioration of the patient's condition and high risk for death. Therefore, Health care providers should be used these patients' characteristics as a warning sign in a patient follow-up to provide early detection and intervention for a favorable outcome. Future researchers shall conduct a prospective study to get a chance to include all important variables like laboratory and radiologic findings.

## Limitation

Due to the retrospective study design, radiographic and laboratory tests-related variables have high missing values and are not included in logistic regression analysis. The effect of the emergency of a new variant virus on the disease outcome is not covered in this study.

## Acknowledgments

We would like to acknowledge Hawassa University, College of Medicine and Health Sciences, the administrative body of both treatment centers from which the data was collected. The data collectors are also going to share gratitude for their contribution to data collection.

## Author Contributions

**Conceptualization:** Samuel Misganaw, Betelhem Eshetu, Adugnaw Adane, Tarekegn Solomon.

**Data curation:** Samuel Misganaw, Betelhem Eshetu, Tarekegn Solomon.

**Formal analysis:** Samuel Misganaw, Betelhem Eshetu, Adugnaw Adane, Tarekegn Solomon.

**Funding acquisition:** Tarekegn Solomon.

**Investigation:** Samuel Misganaw, Adugnaw Adane, Tarekegn Solomon.

**Methodology:** Samuel Misganaw, Betelhem Eshetu, Adugnaw Adane, Tarekegn Solomon.

**Project administration:** Samuel Misganaw.

**Resources:** Samuel Misganaw, Tarekegn Solomon.

**Software:** Samuel Misganaw, Betelhem Eshetu, Adugnaw Adane, Tarekegn Solomon.

**Supervision:** Betelhem Eshetu, Tarekegn Solomon.

**Validation:** Samuel Misganaw, Betelhem Eshetu, Adugnaw Adane, Tarekegn Solomon.

**Visualization:** Samuel Misganaw, Betelhem Eshetu, Adugnaw Adane.

**Writing – original draft:** Samuel Misganaw, Adugnaw Adane, Tarekegn Solomon.

**Writing – review & editing:** Samuel Misganaw, Betelhem Eshetu, Adugnaw Adane, Tarekegn Solomon.

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
