## [Decision Letter · Decision Letter 0]

9 Nov 2022

PONE-D-22-21411Predictors of death among severe COVID-19 patients admitted in Hawassa City, Sidama, Southern Ethiopia: Unmatched case-control studyPLOS ONE

Dear Dr. Misganaw,

Thank you for submitting your manuscript to PLOS ONE. After careful consideration, we feel that it has merit but does not fully meet PLOS ONE’s publication criteria as it currently stands. Therefore, we invite you to submit a revised version of the manuscript that addresses the points raised during the review process. Specifically concerns over the written grammar and spelling need to be addressed before the manuscript could be accepted. It was also stated that the manuscript would benefit if it included a comparison to previously published studies on the subject. 

We look forward to receiving your revised manuscript.

Kind regards,

Colin Johnson, Ph.D.

Academic Editor

PLOS ONE

Journal Requirements:

Reviewers' comments:

Reviewer's Responses to Questions

**Comments to the Author**

1. Is the manuscript technically sound, and do the data support the conclusions?

Reviewer #1: Yes

2. Has the statistical analysis been performed appropriately and rigorously? 

Reviewer #1: Yes

3. Have the authors made all data underlying the findings in their manuscript fully available?

Reviewer #1: Yes

4. Is the manuscript presented in an intelligible fashion and written in standard English?

Reviewer #1: Yes

5. Review Comments to the Author

Reviewer #1: In the study entitled “Predictors of death among severe COVID-19 patients admitted in Hawassa City, Sidama, Southern Ethiopia: Unmatched case-control study” the authors performed a case-control-based study among the severe COVID-19 patients. The aim of the study was to assess the predictors of death among severe COVID-19 patients. The study is good but have some lacunas as listed below:

Comments:

1. The authors should follow the same pattern to represent the numbers. In some places it is written like 5,998,863 whereas in some other places simply 55 000 is written.

2. Lines 68-69 include both the references in the same [].

3. There are lots of formatting issues and grammatical mistakes in the manuscript. For instance in line 71 no space between (chronic kidney disease[25]).

4. The literature survey is not adequately done. A similar perspective (https://doi.org/10.3389/fcimb.2021.674277) was published recently and the authors should compare their results with that of the similar work.

5. Lines 80-81. Please correct the statement “But, majorities of studies on predictor of death and disease severity have been conducted with the context of Asian and American countries”. Please see this article where data from around the world was used (https://doi.org/10.3389/fcimb.2021.674).

6. Line 93, 121 spellings of severe

7. Please check the spellings of severe in full manuscript.

8. Line 225. Gap between 223(59.9%)

9. In some parts of the manuscript, COVID-19 is written whereas in some parts covid-19 is there. Please follow the same notion throughout the manuscript.

6. PLOS authors have the option to publish the peer review history of their article (what does this mean?). If published, this will include your full peer review and any attached files.

Reviewer #1: No

---

## [Author Response · Author response to Decision Letter 0]

5 Dec 2022

Thank you for the comments and information you give to us which will add quality for our manuscript. We tried to incorporate the comments and information given in the revised manuscript. Thank you for your time and considerations.

---

## [Decision Letter · Decision Letter 1]

11 Jan 2023

PONE-D-22-21411R1

Predictors of death among severe COVID-19 patients admitted in Hawassa City, Sidama, Southern Ethiopia: Unmatched case-control study

PLOS ONE

Dear Dr. Misganaw,

Thank you for submitting your manuscript to PLOS ONE. After careful consideration, we feel that it has merit but does not fully meet PLOS ONE’s publication criteria as it currently stands. Therefore, we invite you to submit a revised version of the manuscript that addresses the points raised during the review process.

Of particular note, please address the concerns of the reviewer regarding the literature search in the Introduction, the advancement in knowledge from the study. 

We look forward to receiving your revised manuscript.

Kind regards,

Colin Johnson, Ph.D.

Academic Editor

PLOS ONE

Reviewers' comments:

Reviewer's Responses to Questions

**Comments to the Author**

1. If the authors have adequately addressed your comments raised in a previous round of review and you feel that this manuscript is now acceptable for publication, you may indicate that here to bypass the “Comments to the Author” section, enter your conflict of interest statement in the “Confidential to Editor” section, and submit your "Accept" recommendation.

Reviewer #1: All comments have been addressed

2. Is the manuscript technically sound, and do the data support the conclusions?

Reviewer #1: Yes

3. Has the statistical analysis been performed appropriately and rigorously? 

Reviewer #1: Yes

4. Have the authors made all data underlying the findings in their manuscript fully available?

Reviewer #1: No

5. Is the manuscript presented in an intelligible fashion and written in standard English?

Reviewer #1: No

6. Review Comments to the Author

Reviewer #1: The report entitled “Predictors of death among …….case control study” aims to assess the predictor of death among severely ill COVID-19 patients admitted in Hawassa city. The work is good but suffers from the major drawbacks as follows:

· This is a case control study. The data for control was obtained from May 2021 to June 2021 whereas severe COVID-19 deaths were included from a period of May 2020 to May 2021. There is the disparity of the timelines and the authors should discuss the repercussions of taking the different timelines for the analysis.

· In continuation to the above point the virus has changed a lot during the case-control timelines. The authors should discuss how the mutations in the virus can lead to variations in the results.

The literature search in the introduction section is not adequate. The authors should discuss similar studies including PMID = 34760713.

· It is unclear whether age-matched controls were used in the study.

· It is not clear what knowledge is being added by this study. The conclusions made in the study have already been published by several groups. The authors must discuss any unique points that this study caters to.

· The data was analyzed using SPSS 25. The authors should provide the codes also so that the study can be reproduced.

7. PLOS authors have the option to publish the peer review history of their article (what does this mean?). If published, this will include your full peer review and any attached files.

Reviewer #1: No

---

## [Author Response · Author response to Decision Letter 1]

26 Jan 2023

Thank you for your valuable information and constructive comment. We have corrected the comments . and the statement you wrote well described the availability concern. Others comments are addressed accordingly and incorporated in the revised manuscript.

---

## [Decision Letter · Decision Letter 2]

16 Feb 2023

Predictors of death among severe COVID-19 patients admitted in Hawassa City, Sidama, Southern Ethiopia: Unmatched case-control study

PONE-D-22-21411R2

Dear Dr. Misganaw,

We’re pleased to inform you that your manuscript has been judged scientifically suitable for publication and will be formally accepted for publication once it meets all outstanding technical requirements.

Kind regards,

Colin Johnson, Ph.D.

Academic Editor

PLOS ONE

Additional Editor Comments (optional):

Reviewers' comments:

Reviewer's Responses to Questions

**Comments to the Author**

1. If the authors have adequately addressed your comments raised in a previous round of review and you feel that this manuscript is now acceptable for publication, you may indicate that here to bypass the “Comments to the Author” section, enter your conflict of interest statement in the “Confidential to Editor” section, and submit your "Accept" recommendation.

Reviewer #1: All comments have been addressed

2. Is the manuscript technically sound, and do the data support the conclusions?

Reviewer #1: Yes

3. Has the statistical analysis been performed appropriately and rigorously? 

Reviewer #1: Yes

4. Have the authors made all data underlying the findings in their manuscript fully available?

Reviewer #1: No

5. Is the manuscript presented in an intelligible fashion and written in standard English?

Reviewer #1: Yes

6. Review Comments to the Author

Reviewer #1: All the concerns are addressed. The manuscript can be accepted for the publication in the current format.

7. PLOS authors have the option to publish the peer review history of their article (what does this mean?). If published, this will include your full peer review and any attached files.

Reviewer #1: No

---

## [Editor Report · Acceptance letter]

20 Feb 2023

PONE-D-22-21411R2 

Predictors of death among severe COVID-19 patients admitted in Hawassa City, Sidama, Southern Ethiopia: Unmatched case-control study 

Dear Dr. Misganaw:

I'm pleased to inform you that your manuscript has been deemed suitable for publication in PLOS ONE. Congratulations! Your manuscript is now with our production department. 

Kind regards, 

on behalf of

Dr. Colin Johnson 

Academic Editor

PLOS ONE